# Effects of Anti-Pseudomonal Agents, Individually and in Combination, With or Without Clarithromycin, on Growth and Biofilm Formation by Antibiotic-Susceptible and -Resistant Strains of *Pseudomonas aeruginosa,* and the Impact of Exposure to Cigarette Smoke Condensate

**DOI:** 10.3390/antibiotics14030325

**Published:** 2025-03-19

**Authors:** Moloko C. Cholo, Charles Feldman, Ronald Anderson, Lebogang Sekalo, Naledi Moloko, Guy A. Richards

**Affiliations:** 1Department of Immunology, Faculty of Health Sciences, University of Pretoria, Pretoria 0001, South Africanaledi.moloko@tuks.co.za (N.M.); 2Basic and Translational Research Unit, Nuclear Medicine Research Infrastructure, Steve Biko Academic Hospital, Pretoria 0001, South Africa; 3Department of Internal Medicine, Faculty of Health Sciences, University of the Witwatersrand, Johannesburg 2193, South Africa; charles.feldman1@wits.ac.za; 4Department of Surgery, Division of Critical Care, Faculty of Health Sciences, University of the Witwatersrand, Johannesburg 2193, South Africa

**Keywords:** amikacin, cefepime, ciprofloxacin, clarithromycin, biofilm, cigarette smoke, condensate, fractional inhibitory concentration index, minimal inhibitory concentration, *Pseudomonas aeruginosa*, synergy

## Abstract

**Background/Objectives:** *Pseudomonas aeruginosa* (*Psa*) can circumvent antimicrobial chemotherapy, an ability enhanced by cigarette smoking (CS). This study probed potential benefits of combinations of anti-pseudomonal agents, and potential augmentation by a macrolide, in the absence or presence of cigarette smoke condensate (CSC). **Methods:** Two susceptible (WT: wild-type and DS: drug-sensitive) and one multidrug-resistant (MDR) strains of *Psa* were treated with amikacin, cefepime, and ciprofloxacin, individually and in combination, and with and without clarithromycin, followed by the measurement of planktonic growth and biofilm formation by spectrophotometry. Antibiotic interactions were determined using the fractional inhibitory concentration index (FICI) method. Effects on preformed biofilm density were measured following the addition of antibiotics: all procedures were performed in the absence and presence of CSC. **Results:** The minimal inhibitory concentrations (MICs) of the three agents ranged from 0.125 mg/L to 1 mg/L (WT and DS strains) and 16 mg/L to 64 mg/L (MDR strain), with all resistant to clarithromycin (125 mg/L). MIC values closely correlated with the antibiotic concentrations required to inhibit biofilm formation. FICI revealed synergism between most combinations, with augmentation by clarithromycin. Amikacin had the greatest effect on biofilm density, which was potentiated by combination with the other antibiotics, particularly clarithromycin. Exposure to CSC had variable, albeit modest, effects on bacterial growth and biofilm formation, but low concentrations increased biofilm mass and attenuated synergistic antimicrobial interactions and effects on biofilm density. **Conclusions:** Amikacin, cefepime, and ciprofloxacin, especially with clarithromycin, exhibit synergistic anti-pseudomonal activity and decrease preformed biofilm density. CSC attenuated these effects, illustrating the pro-infective potential of CS.

## 1. Introduction

*Pseudomonas aeruginosa* (*Psa*) is a ubiquitous Gram-negative bacterium, found in common abiotic environments, such as soil and water, and on and in living organisms [1]. In humans, it frequently colonises the compromised respiratory tract and is an important nosocomial pathogen, particularly in the intensive care unit (ICU), where patients are frequently mechanically ventilated, have multiple invasive monitoring devices and catheters, have been receiving broad-spectrum antibiotics, and may be immunocompromised [2,3,4,5,6].

*Psa* incorporates a variety of virulence factors, which include the lipopolysaccharide outer membrane, motility determinants (flagelli, type IV pili), protein secretion systems (exotoxin A, proteolytic enzymes, phospholipase C), and the pyocyanin metabolite [4,7]. In addition, it incorporates systems essential for metabolism, transportation, and efflux of cellular materials [2,3,5].

Similar to other bacteria, *Psa* has mechanisms that allow it to adapt to environmental changes [3,5], which include its ability to develop biofilms, that provide resistance to environmental stressors such as antibiotics, disinfectants, and host immune defences, and promote the persistence of the microorganism [4] in the environment or lungs for several months or years [7,8]. Biofilms are highly structured, aggregated communities of sessile bacterial cells, encased within an extracellular polymeric substance (EPS) or matrix (ECM) that adheres to abiotic or biological surfaces [2], serving as a physical barrier, and rendering sessile bacteria (slow-growing, persisters and non-metabolic, dormant cells) tolerant to antibiotic treatment [3,4,9].

The antibacterial armamentarium against *Psa* is limited due to the acquisition of resistance determinants by the pathogen, but includes aminoglycosides, fluoroquinolones, polymixins, and anti-pseudomonal beta (β)-lactams [3]. The microorganism has been included in the World Health Organisation’s (WHO) list of priority pathogens, for which new antimicrobials need to be urgently developed because of its extensive resistance to several classes of antibiotics [2,3,4,5,10].

Patients with chronic *Psa* respiratory tract infections have been successfully treated with combinations of macrolide antibiotics together with anti-pseudomonal agents, because the former, although not lethal for *Psa* [11,12,13], can inhibit quorum sensing, swarming, twitching, biofilm formation, bacterial colonisation, regrowth and production of virulence factors such as proteases, phospholipase C [14,15,16] and pyocyanin pigment [17,18,19,20,21,22], as well as decrease the density of preformed biofilm [23].

Cigarette smoke (CS) exposure is a risk factor for the development of pulmonary *Psa* infection in smokers and those with chronic structural lung disease [24,25,26], promoting bacterial colonisation and increasing bacterial load [26,27]. CS increases bacterial resistance to killing by polymorphonuclear leukocytes (PMNL), by increasing resistance to oxidative stress [25,28], increasing the virulence of *Psa* as demonstrated by the increased mortality of mice infected with the organism [25], and by increasing biofilm formation [25,28,29]. CS condensate (CSC) may induce antibiotic resistance in *Psa* [30] by upregulation of genes encoding multidrug efflux pumps, via suppression of the OprD porin involved in carbapenem influx, and through other genes involved in resistance expression [28,30,31,32].

Despite these effects on anti-pseudomonal agents, limited information is available on the effects of CSC on anti-pseudomonal antibiotic combinations and potentiation by macrolides. This laboratory study probed the potential benefit of combinations of key anti-pseudomonal agents augmented by the inclusion of clarithromycin. These antibiotics other than clarithromycin are classical anti-pseudomonal agents, and as such are ideal agents to evaluate in this study, firstly, against drug-sensitive organisms, and then, against more resistant organisms. While clarithromycin is not an anti-pseudomonal agent, it is, along with other macrolides, an agent that can inhibit quorum sensing and as such can inhibit biofilm formation without having any specific microbicidal activity. All experimental procedures were performed in the absence and presence of CSC. A summary of the experimental procedures performed on the three strains of *Psa* is shown in Figure 1.

## 2. Results

### 2.1. Bacterial Inoculum

The cultures of the three *Psa* strains harvested for inoculum preparation were grown to optical density (OD) readings of 0.959 ± 0.066, 0.977 ± 0.189, and 0.962 ± 0.046 at 600 nm for the wild-type (WT), drug-sensitive (DS), and multidrug-resistant (MDR) strains, respectively. The numbers of bacteria in the assays of these three strains were comparable and were 8.82 × 10^4^ (±2.53 × 10^4^), 9.5 × 10^4^ (±1.44 × 10^4^), and 8.42 × 10^4^ (±1.85 × 10^4^) colony-forming units (cfu)/mL, respectively.

### 2.2. Bacterial Growth Rates

The rates of growth and biofilm formation of the DS and the MDR *Psa* strains were compared with those of the WT strains at each time point (Figure 2). In planktonic cultures, the WT and DS strains achieved optimum growth at 12 h with the DS strain growing faster at this time point (*p* < 0.05). The MDR strain showed slower growth rates early (<12 h) (*p* < 0.05) but surpassed the WT at 18 h (*p* < 0.05) (Figure 2a). The *p* values at 0, 6, 12, 18, and 24 h were 0.945, 0.8, <0.0001, 0.0497, and 0.0002 for the DS strain and 0.43, 0.118, for 0 and 6 h and <0.0001 for 12, 18, and 24 h for the MDR strain, respectively.

For biofilm-forming cultures, all strains achieved optimum biofilm formation at 12 h. However, rates of biofilm formation by the DS and MDR strains were significantly slower than that of the WT strain (*p* < 0.05) (Figure 2b). No *p* values were determined at 0 hr for all the strains as there was no biofilm developed. The *p* values at 12, 24, 36, and 24 h were all <0.0002 between the WT and the DS strain, while they were <0.0063, 0.0117, 0.0070, and 0.0002, respectively, between the WT and the MDR strains.

### 2.3. Minimum Inhibitory Concentration (MIC) Determinations for the Pseudomonas aeruginosa Strains

In planktonic cultures, the MICs of the anti-pseudomonal agents for the WT and DS strains were low (0.125–1 mg/L), while those for the MDR strain were high (16–64 mg/L) (Table 1a). However, in the case of clarithromycin, the MIC for all three strains was high (125 mg/L).

In the biofilm-forming cultures, as expected, the inhibitory effects of the antibiotics on biofilm formation coincided closely with the MIC values of the antibiotics on planktonic cultures of all three strains of the pathogen.

In the case of the preformed biofilm (Table 1b), no antibiotic, at any concentration, achieved complete eradication of the biofilm. Amikacin was the most active, attaining a ≥ 50% decrease in biofilm density at concentrations of 32 mg/L for the WT strain and 64 mg/L for the DS and MDR strains. Clarithromycin demonstrated activity only against the preformed biofilm of the WT strain, achieving a 21% reduction in density compared to the controls.

### 2.4. Effects of Combinations of the Anti-Pseudomonal Agents and Clarithromycin Against Planktonic [and Biofilm-Forming] Cultures and Preformed Biofilm Cultures [Biofilm Degradation]

These results are shown in Table 2. A total of eleven antibiotic combinations were evaluated against each strain, consisting of six two-drug, four three-drug, and one four-drug combination(s). In planktonic cultures (Table 2a), all combinations showed synergy with the WT. For the DS strain, all the two- and three-drug combinations showed synergy, with an additive effect with four-drugs. For the MDR strain, three sets of two-drug (clarithromycin [CLA] + cefepime [CEF], CLA + ciprofloxacin [CIP], and amikacin [AMI] + CEF) and three of the three-drug combinations (CLA + AMI + CEF, CLA + AMI + CIP, and CLA + CEF + CIP) were synergistic, while the remaining five antibiotic combinations (CLA + AMI, CEF + CIP, AMI + CIP, AMI + CEF + CIP, and a four-drug combination) demonstrated additive effects. No combination showed indifferent or antagonistic effects against the planktonic growth of any bacterial strain. The fractional inhibitory concentration index (FICI) values for synergism ranged between 0.125 and 0.5 for the WT and DS strains, while those for the MDR strain were 0.5, demonstrating minimal synergy. Similar to the MIC determinations, the effects of the antibiotic combinations on biofilm formation were similar to those of the planktonic cultures. 

Several antibiotic combinations were highly active against the preformed biofilm of the DS strain, with six combinations demonstrating a ≥50% reduction in biofilm density relative to the non-antibiotic-treated controls (Table 2b). Four combinations (CLA + CIP, CLA + AMI + CIP, AMI + CEF + CIP, and a four-drug combination) demonstrated synergy. The most effective synergistic combinations included ciprofloxacin (four of the above combinations) and amikacin (three combinations). Inclusion of clarithromycin further augmented the synergistic interactions. With regard to the WT and MDR strains, no combination achieved ≥50% decrease in the density of preformed biofilm. Although possibly not the sole reason, the greater sensitivity (lower MIC) of the DS strain is probably responsible for less preformed biofilm produced by this strain relative to the other strains.

### 2.5. Effect of Cigarette Smoke Condensate (CSC) on Bacterial Growth and Biofilm Formation

In planktonic cultures, exposure of the three bacterial strains to varying concentrations of CSC showed different effects on bacterial growth (Appendix A). For the WT strain, there was a dose–response relationship with increasing doses of CSC augmenting bacterial growth, with statistical significance at concentrations of 3.125 mg/L (*p* < 0.05). However, although not statistically significant, further exposure to higher concentrations demonstrated an opposite response with a decrease in bacterial growth. There was a similar dose–response relationship with the DS and MDR strains in which CSC increased bacterial growth throughout all concentrations, with statistical significance at ≥100 mg/L and 12.5 mg/L, respectively. In contrast to the WT, no CSC concentration resulted in inhibition in bacterial growth of these other two strains.

In biofilm-forming cultures, CSC exposure increased biofilm formation for all three strains (Appendix A). At lower concentrations of CSC, WT cultures increased biofilm formation with statistical significance evident at 25 mg/L. However, higher concentrations caused a non-significant decrease. This latter pattern was similar to that of the DS strain, although these effects were not significant at any concentration. Conversely, exposure to CSC increased biofilm formation by the MDR strain at all concentrations in a dose-dependent manner, achieving statistical significance at 400 mg/L (*p* < 0.05).

With regard to the effect on preformed biofilm density, only the WT strain was treated with CSC (Appendix A). Exposure to lower CSC concentrations (0.4–12.5 mg/L) was associated with a statistically significant increase in biofilm density (*p* < 0.05), whereas at higher concentrations, density decreased, achieving statistical significance at ≥100 mg/L.

### 2.6. MIC Determination in the Presence of CSC

Determination of the MICs of the antibiotics was performed in the presence of 3.125 mg/L of CSC, the concentration that had significantly increased the growth of the WT strain (Table 3). In planktonic and biofilm-forming cultures of all three bacterial strains, addition of CSC did not alter the MICs (Table 3a). For preformed biofilm, exposure to CSC reduced the inhibitory activity of the individual antibiotics, resulting in increased residual biofilm density (Table 3b).

### 2.7. Effect of CSC on Antibiotic Combinations

As with the MIC determinations, FICI was also determined in the presence of a CSC concentration of 3.125 mg/L (Table 4). For planktonic and biofilm-forming cultures, exposure of all three strains to CSC did not influence the efficacy of the antibiotic combinations with the FICI values unchanged for all combinations (Table 4a). In contrast, the reduction in density of preformed biofilm achieved by the combinations was attenuated by CSC in all three strains (Table 4b).

## 3. Discussion

*Psa* is a common cause of healthcare-associated infections in critically ill and immunocompromised patients. Due to high levels of antibiotic resistance to currently available anti-pseudomonal agents, empiric antibiotic therapy is often inappropriate, and outcomes are poor. The addition of a macrolide to therapy can enhance the activity of those agents to which the organism is sensitive, by inhibiting quorum sensing and the formation of biofilm [14,15,16]. In contrast, exposure to CS has the potential to exacerbate *Psa* infections, promoting bacterial growth and biofilm formation and increasing antibiotic resistance [28,30,31].

Despite these reports, the impact of CS exposure on the antimicrobial activity of anti-pseudomonal agents, in combination with a macrolide, has not previously been reported. In the current study, we investigated the effect of CSC on the antimicrobial activity of three anti-pseudomonal agents alone and in various combinations, with or without clarithromycin, against three *Psa* strains in vitro [33,34,35,36]. Amikacin, cefepime, ciprofloxacin, and clarithromycin were first evaluated to determine their individual MIC values and thereafter, they were tested in various combinations to assess their effects on planktonic growth, biofilm-formation, and the density of preformed biofilm of all three pseudomonas strains.

These strains differed as to their properties. Whereas growth was optimally achieved at similar rates at 12 h for the WT and DS strains, the MDR strain had a slower growth rate in planktonic cultures. However, the DS and MDR strains produced less biofilm than did the WT. Furthermore, biofilm formation was shown to be independent of the susceptibility profile of the bacterial strains. Despite these differences, these three *Psa* strains produced biofilm optimally at 12 h in in vitro cultures.

As expected, the three strains demonstrated different responses to the anti-pseudomonal agents, with the MIC values of the drugs being lower in the WT and DS strains compared with the MDR strain against which these agents would not be expected to be active. The differences in the MICs of the antibiotics to the individual strains were similar in the planktonic and biofilm-forming cultures.

We evaluated FICI values with a value ≤ 0.5 representing synergy. Lower but similar FICI values were achieved for the planktonic and biofilm-forming cultures of the WT and DS strains, ranging from 0.125 to 0.5, while those of the MDR strain were higher, mostly at 0.5, and were comparable between the planktonic and biofilm-forming cultures.

For clarithromycin, the inhibitory antimicrobial activity was similar among the three strains and was comparable for the planktonic and biofilm-forming cultures. The degree of inhibition demonstrated by the individual antibiotics against preformed biofilm was, however, comparable among the three strains, irrespective of the susceptibility profile of the strain. The failure of the antibiotics to decrease the density of preformed biofilm completely, illustrates the inaccessibility of antibiotics to the organisms in biofilm, promoting failure of eradication and persistence of disease. Despite this, of the agents used, amikacin and to a lesser extent ciprofloxacin, both singly and in combination, demonstrated greater, but nevertheless incomplete inhibitory activity against preformed biofilm. In contrast, although inactive against preformed biofilm on its own, clarithromycin demonstrated high inhibitory activity when used in combination with amikacin and ciprofloxacin, as demonstrated in three of the four combinations, with synergy against the DS strain in particular (>50% reduction in the density of preformed biofilm). These results demonstrated that while low concentrations (achievable MICs) of antibiotics to which the organism is sensitive are required to inhibit bacterial growth and biofilm formation, higher concentrations are required to decrease the density of preformed biofilm.

Exposure to CSC increased bacterial growth and production of preformed biofilm, which confirmed the findings of other studies, including those that studied *Psa* and other bacterial species such as *Mycobacterium tuberculosis*, *Staphylococcus aureus*, *Streptococcal species*, and *Bifidobacterium animalis* [37,38,39,40,41]. Interestingly, these findings demonstrated that lower CSC concentrations are necessary for promoting bacterial growth (3.125–6.25 mg/L) and potentiating biofilm formation (0.4–12.5 mg/L) in the WT strain, while in contrast, higher CSC concentrations (12–400 mg/L) are required to achieve similar effects in clinical isolates, particularly the MDR strain, highlighting the varying responses of the different bacterial strains. Future studies involving multiple *Psa* strains are necessary to determine the implications of these findings.

CSC exposure did not, however, impact the MICs of the anti-pseudomonal agents, either alone or in combination, against the planktonic and biofilm-forming cultures, despite promoting bacterial growth and biofilm formation. In contrast, previous studies have shown that CSC exposure increases the MICs of certain antibiotics such as levofloxacin, imipenem, tigecycline, and minocycline against *Psa* [25,30] and has also increased the expression of antibiotic resistance genes such as the *mexA/X/Z* and *mexEF-oprN* multidrug efflux pumps and the *nfxC* gene that encodes for fluoroquinolone resistance [30,32], while suppressing the *oprD* porin gene allowing carbapenem access to the penicillin-binding proteins [28,30,31]. The failure to demonstrate an effect on the antimicrobial activity of the antibiotics in the current study may have been due to the different experimental approaches used in other studies, for example, the length of exposure to CSC, or that different antibiotics were evaluated. However, in the current study, CSC exposure did demonstrate a reduction in the capacity of the antibiotics singly and in combination to reduce the density of preformed biofilm by all three strains.

Antibiotics would be less effective against the eradication of pseudomonas in cigarette smokers as their ability to disperse preformed biofilm would be further compromised. This indicates that smokers with pseudomonal infections (and potentially other organisms) of the respiratory tract would be less likely to respond unless therapy is initiated promptly.

In summary, the anti-pseudomonal agents, singly and in combination and with or without clarithromycin, were effective against planktonic and biofilm-forming cultures, in the absence or presence of CS exposure, and the effect was comparable between these two cultures. However, the antibiotics had limited effect on preformed biofilm density, and this was further reduced by exposure of the bacteria to CSC. However, of the antibiotics, amikacin and ciprofloxacin were the most efficacious against preformed biofilm, and this was potentiated by clarithromycin.

## 4. Materials and Methods

### 4.1. Strains and Growth Media

Three *Psa* bacterial strains were used. These included a reference strain (PA01) and a clinical isolate, both of which were DS, and an MDR clinical isolate. These were provided pro deo by the National Institute of Communicable Diseases (NICD), of the National Health Laboratory Service (NHLS), South Africa, upon obtaining an access permit.

The culture media used included Luria-Bertani (LB) broth (pH 7.5), 10% tryptic soy broth (TSB, pH 7.5), and Luria agar (LA) (Lasec, Johannesburg, South Africa). The LA medium was used for the development of colonies, which were utilised for preparation of numerically standardised bacterial suspensions. These were subsequently inoculated into the LB and 10% TSB broths for preparation of planktonic and biofilm-promoting cultures, respectively.

### 4.2. Antibiotics, CSC, and Chemicals

The antibiotics used were amikacin, ciprofloxacin, and cefepime, with clathrithromycin as the sole macrolide. All were purchased from Sigma-Aldrich (St. Louis, MO, USA). Amikacin was dissolved in sterile distilled water, ciprofloxacin in 0.1% hydrochloric acid (HCl), and cefepime and clarithromycin were dissolved in dimethylsulfoxide (DMSO). The CSC was purchased from Murty Pharmaceuticals (Lexington, KY, USA) and prepared as a concentrate in DMSO. Unless otherwise indicated, all chemicals used were purchased from Sigma-Aldrich (St. Louis, MO, USA), Whitehead Scientific, and Lasec (Johannesburg, South Africa).

### 4.3. Preparation of Bacterial Inoculum

Ten microlitres of frozen stock of each bacterial strain was inoculated onto an LA agar plate and the plate was incubated for 24 h for colony development. One colony was inoculated into 10 mL of LB broth and incubated for 24 h at 37 °C at 180 rotations per minute (rpm) and allowed to grow to an OD of 1 at 600 nm. The culture was adjusted with LB broth to OD_600_ = 0.25–0.28, which yields approximately 10^7^ cfu/mL bacterial numbers in each inoculum: determined using dilution theory [1,33,42,43,44].

### 4.4. Preparation of Planktonic and Biofilm-Forming Cultures

The cultures were prepared as previously described with minor modifications [1,43]. Briefly, for planktonic cultures, 10^5^ cfu/mL from the prepared inoculum (approximately 10^7^ cfu/mL) were added to LB broth in 96-well plates to final volumes of 200 µL. The cultures were thoroughly mixed and incubated at 37 °C for 24 h. Bacterial growth was determined spectrophotometrically (Powerwave_x_ spectrophotometer, Bio-Tek Instrument, Inc., Winooski, VT, USA) at 600 nm. The ODs of the cultures were measured at the initial time point, referred to as Day zero (D0) and the final time point at 24 h, referred to as Day one (D1). Bacterial growth was determined by an increase in OD readings between D0 and D1.

For preparation of biofilm forming cultures, the same numbers of bacteria used for the planktonic cultures were added to 10% TSB growth medium in 96-well plates to final volumes of 200 µL. The cultures were thoroughly mixed, then wrapped with parafilm and incubated at 37 °C for 48 h without shaking, to allow for biofilm development. Biofilm growth was determined at the final time point (48 h) (Day two: D2) in each well by the quantification method described below.

### 4.5. Biofilm Quantification

Biofilm formation was determined using the crystal violet staining procedure with minor modifications [1,44]. The planktonic cells, together with the unattached cells and the growth medium were removed from the wells by aspiration and the wells washed with 200 µL sterile distilled water. The adherent cells with biofilm were stained by adding 200 µL of crystal violet (0.001% weight per volume (*w*/*v*) in water) and incubated for 1 hr at room temperature. Excess crystal violet dye was removed by aspiration and the wells washed once with 200 µL of water and the plates air-dried at room temperature. Biofilm was extracted by adding 200 µL of 70% ethanol into each well followed by incubation for 30 min at room temperature. The contents of each well containing the crystal violet-ethanol solution were measured by OD at 570 nm. The amount of crystal violet detected colourimetrically was recorded and any increase relative to the D0 reading was considered to represent biofilm formation.

### 4.6. Determination of Bacterial Growth Rates

To determine the rates of bacterial growth and biofilm development, separate sets of planktonic and biofilm-forming cultures were prepared and incubated at 37 °C as described above for different final time points. The final time points were 0, 6, 12, 18, and 24 h for planktonic and 0, 12, 24, 36, and 48 h for biofilm-forming cultures.

### 4.7. Antibiotic Activities Using MIC Determination

The antimicrobial activity of the anti-pseudomonal agents and clarithromycin were evaluated individually by MIC determination. For planktonic and biofilm-forming cultures, 2 µL volumes of the anti-pseudomonal agents were prepared in double-dilutions to yield the following final concentrations: amikacin 0.06–8 mg/L, cefepime 0.015–2 mg/L, ciprofloxacin 0.006–1 mg/L, and clarithromycin 2–250 mg/L were added to one set of wells and the solvents to the control wells. To study the effects on biofilm formation, the antibiotics were initially added to the biofilm-forming cultures at final concentrations of 2–64 mg/L for the anti-pseudomonal agents and 2–250 mg/L for clarithromycin and incubated for 48 h, after which biofilm formation was measured and compared with controls. To determine the effects on the density of established biofilm, the antibiotics were added after 48 h of incubation and thereafter incubated for a further 48 h after which biofilm formation was again measured and compared with that in the original experiments. For all the cultures, the lowest concentrations of the individual antibiotics that showed ≥50% inhibition of planktonic growth, biofilm development or decrease in biofilm density were used for the MIC determinations.

### 4.8. FICI Determination of Synergy

The antibiotic mixtures were prepared in various combinations in three sub-MICs, specifically 0.5, 0.3, and 0.25 of their MIC values for the two-, three- and four-drug combinations, respectively. Combinations of the anti-pseudomonal agents with clarithromycin were evaluated by determining the FICI according to the Loewe Additivity Model as previously described [33]. The FICI was calculated as the sum of the fractional inhibitory concentrations (FICs) of the individual antibiotics, where the FIC was determined as the MIC of the antibiotic in each combination relative to the MIC of the antibiotic alone. The inhibitory interaction of the antibiotics was synergistic, additive, indifferent or antagonistic when the FICI value was ≤0.5, 0.5–1.0, 1.0–4.0 or >4.0, respectively.

### 4.9. Effects of CSC

Two microlitres of varying concentrations of CSC (0.4–400 mg/L) were added to each CSC-treated well, and DMSO alone (1%, final) was added to the control wells. The cultures were incubated, and growth and biofilm formation were measured as described above.

In order to assess the effects of CSC on the sustained production of preformed biofilm, various concentrations of CSC were added to the individual culture wells selected for CSC treatment. The cultures were incubated, and biofilm quantification performed as described above with the results expressed as the percentage reduction in biofilm density.

With respect to effects of CSC on antibiotic activity, the bacterial cultures were prepared and a fixed dose (mg/L) of CSC was added to all the wells and the cultures pre-incubated for 5–10 min at room temperature. Thereafter, the various concentrations of antibiotics singly, and in combination, were added and the cultures incubated, followed by determinations of MIC and FICI for growth, biofilm formation, and change in biofilm density as mentioned above.

### 4.10. Statistical Analysis

For each series, three biological replicates were conducted in three technical replicates. The results for each strain of *Psa* were expressed as mean values ± standard deviations (SDs) and comparisons between anti-pseudomonal agents/clarithromycin-treated and-untreated and CSC-unexposed and exposed, and anti-pseudomonal agents with clarithromycin-treated and -untreated cultures were performed using the unpaired Student *t*-test/Mann–Whitney U-test with a *p* value of <0.05 taken as being statistically significant.

## 5. Conclusions

In conclusion, these results have illustrated that the tested anti-pseudomonal agents alone and in combination with clarithromycin would be effective in treating planktonic and biofilm-forming cultures of sensitive organisms, even in the presence of CSC. In contrast, the ability of these antibiotics to decrease the density of preformed biofilm singly and in combination was inefficient, even with high concentrations, and this was exacerbated by exposure to CSC. Although clarithromycin, per se, had poor antibiofilm activity against preformed biofilm, it demonstrated inhibitory activity when used in combination with certain of the anti-pseudomonal agents tested, specifically, ciprofloxacin and amikacin. These findings suggest that prompt treatment of an infection would be more effective prior to the development of biofilm. Antibiotic combinations comprising clarithromycin and an active anti-pseudomonal agent, such as amikacin and ciprofloxacin would be effective regimens for the treatment of sensitive *Psa* infections.

## Figures and Tables

**Figure 1 antibiotics-14-00325-f001:**
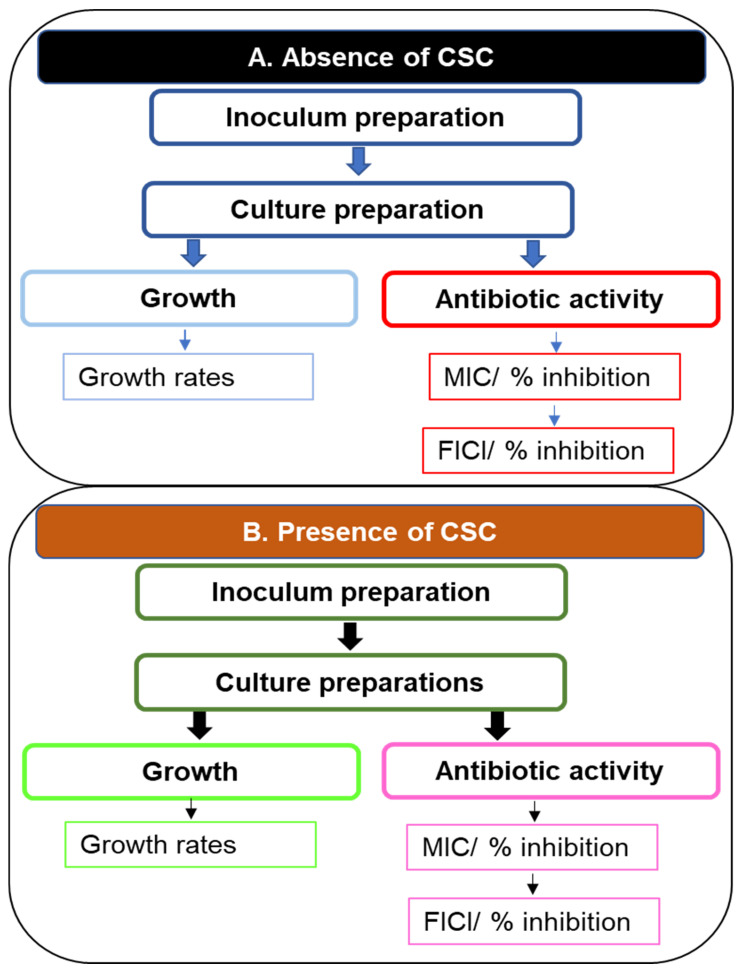
A flow diagram illustrating experiments performed on cultures of three strains of *Psa*: An inoculum was prepared for each strain, followed by preparation of different cultures to evaluate planktonic growth or biofilm formation by the organism. Antibiotic activity was determined by establishing the MICs of individual antibiotics or combinations by means of the fractional inhibitory FICI. All experiments were performed in the absence and presence of CSC. Abbreviations: CSC, cigarette smoke condensate; FICI, fractional inhibitory concentration index; MIC, minimum inhibitory concentration; *Psa*, *Pseudomonas aeruginosa*.

**Figure 2 antibiotics-14-00325-f002:**
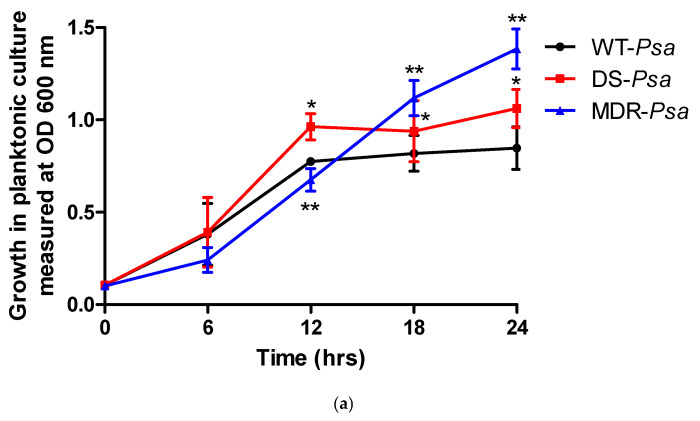
The rates of (**a**) bacterial growth and (**b**) biofilm formation of the *Pseudomonas aeruginosa* (*Psa*) strains compared with wild-type (WT) at each time point with statistical significance at a *p* value ≤ 0.05 represented as (*) for WT versus drug-sensitive (DS) and (**) for WT versus multidrug-resistant (MDR) strains. Abbreviations: DS, drug-sensitive; MDR, multidrug-resistant; OD, optical density; *Psa*, *Pseudomonas aeruginosa*; WT, wild-type.

**Table 1 antibiotics-14-00325-t001:** Individual minimum inhibitory concentrations (MICs) of clarithromycin and the anti-pseudomonal agents against *Pseudomonas aeruginosa* (*Psa*), and their effects on biofilm, in (**a**) planktonic growth [and biofilm-forming cultures] and (**b**) preformed biofilm cultures [biofilm degradation] in the absence of cigarette smoke condensate (CSC).

(a)				(b)		
MIC	MIC in mg/L (% Inhibition) in Planktonic Growth [and Biofilm-Forming Cultures] in the Absence of CSC			MIC in mg/L (% Decrease in Density of Biofilm) in Preformed Biofilm Cultures in the Absence of CSC		
Strain	WT	DS	MDR	WT	DS	MDR
Clarithromycin	125 (51%)	125 (54%)	125 (58%)	125 (21%)	>250 (no inhibition)	>250 (no inhibition)
Amikacin	1 (59%)	1 (72%)	64 (62%)	32 (53%)	64 (52%)	64 (62%)
Cefepime	0.5 (63%)	1 (77%)	32 (71%)	64 (24%)	64 (32%)	64 (14%)
Ciprofloxacin	0.25 (60%)	0.125 (61%)	16 (70%)	64 (33%)	64 (42%)	64 (38%)

Abbreviations: CSC, cigarette smoke condensate; DS, drug-sensitive; MDR, multidrug-resistant; MIC, minimum inhibitory concentration, WT, wild-type. Yellow shading, >50% reduction in density of preformed biofilm.

**Table 2 antibiotics-14-00325-t002:** Fractional inhibitory concentration index (FICI) values of combinations of clarithromycin and the anti-pseudomonal agents against *Pseudomonas aeruginosa* (*Psa*) strains in (**a**) planktonic growth [and biofilm-forming cultures] and (**b**) preformed biofilm cultures in the absence of cigarette smoke condensate (CSC).

(a)				(b)		
FICI	FICI (% Inhibition) in Planktonic Growth [and Biofilm-Forming Cultures] in the Absence of CSC			FICI (% Decrease in Density of Biofilm) in the Preformed Biofilm Cultures in the Absence of CSC		
Strains	WT	DS	MDR	WT	DS	MDR
CLA + AMI	0.5 (85%)	1 (70%)	1 (67%)	1 (9%)	1 (60%)	1 (9%)
CLA + CEF	0.25 (69%)	0.5 (92%)	0.5 (76%)	>1 (no inhibition)	>1 (no inhibition)	>1 (no inhibition)
CLA + CIP	0.5 (66%)	0.5 (91%)	0.5 (82%)	0.5 (25%)	0.015 (56%)	0.25 (15%)
AMI + CEF	0.25 (76%)	0.25 (89%)	0.5 (92%)	1 (12%)	>1	1 (18%)
AMI + CIP	0.5 (75%)	0.25 (69%)	1 (89%)	0.5 (32%)	0.5 (24%)	1 (20%)
CEF + CIP	0.5 (68%)	0.25 (91%)	1 (92%)	0.5 (28%)	1 (50%)	1 (8%)
CLA + AMI + CEF	0.25 (74%)	0.5 (91%)	1 (90%)	1 (5%)	0.125 (31%)	>1 (no inhibition)
CLA + AMI + CIP	0.5 (73%)	0.5 (92%)	0.5 (67%)	1 (25%)	0.015 (50%)	0.25 (14%)
CLA + CEF + CIP	0.25 (71%)	0.5 (92%)	0.5 (76%)	>1 (no inhibition)	1 (35%)	>1 (no inhibition)
AMI + CEF + CIP	0.25 (77%)	0.25 (92%)	1 (91%)	0.25 (21%)	0.03 (50%)	1 (9%)
All four drugs	0.25 (71%)	1 (89%)	1 (91%)	0.5 (20%)	0.03 (63%)	0.5 (12%)

Abbreviations: AMI, amikacin; CEF, cefepime; CIP, ciprofloxacin; CLA, clarithromycin; CSC, cigarette smoke condensate; DS, drug-sensitive; FICI, fractional inhibitory concentration index; MDR, multidrug-resistant; WT, wild-type. Yellow shading, >50% reduction in density of preformed biofilm.

**Table 3 antibiotics-14-00325-t003:** Minimum inhibitory concentrations (MICs) of clarithromycin and the anti-pseudomonal agents against the strains of *Pseudomonas aeruginosa* (*Psa*) in (**a**) planktonic growth [and biofilm-formingcultures] and (**b**) preformed biofilm cultures in the presence of cigarette smoke condensate (CSC).

(a)				(b)		
**MIC**	MIC in mg/L (% Inhibition) in Planktonic Growth [and Biofilm-Forming Cultures] in the Presence of CSC			MIC in mg/L (% Decrease in Density of Preformed Biofilm Cultures in the Presence of CSC)		
Strain	WT CSC	DS CSC	MDR CSC	WT CSC	DS CSC	MDR CSC
Clarithromycin	125 (54%)	125 (53%)	125 (54%)	125 (18%)	>250 (no inhibition)	>250 (no inhibition)
Amikacin	1 (59%)	1 (80%)	64 (63%)	32 (51%)	64 (51%)	64 (60%)
Cefepime	0.5 (59%)	1 (62%)	32 (75%)	64 (17%)	64 (29%)	>64 (no inhibition)
Ciprofloxacin	0.25 (51%)	0.125 (70%)	16 (65%)	64 (37%)	64 (44%)	128 (42%)

Abbreviations: CSC, cigarette smoke condensate; DS, drug-sensitive; MDR, multidrug-resistant; MIC, minimum inhibitory concentration; WT, wild-type. Yellow shading, >50% reduction in biofilm quantity.

**Table 4 antibiotics-14-00325-t004:** Fractional inhibitory concentration index (FICI) values of combinations of clarithromycin and the anti-pseudomonal agents against the *Pseudomonas aeruginosa* (*Psa*) strains in the (**a**) planktonic growth [and biofilm-forming cultures] and (**b**) preformed biofilm cultures in the presence of cigarette smoke condensate (CSC).

(a)				(b)		
FICI	FICI Values (% Inhibition) in Planktonic Growth [Biofilm-Forming Cultures] in the Presence of CSC			FICI Values (% Inhibition) in Preformed Biofilm Cultures in the Presence of CSC		
Strains	WT CSC	DS CSC	MDR CSC	WT CSC	DS CSC	MDR CSC
CLA + AMI	0.5 (73%)	1 (72%)	1 (65%)	>1 (no inhibition)	1 (58%)	>1 (no inhibition)
CLA + CEF	0.25 (68%)	0.5 (91%)	0.5 (76%)	>1 (no inhibition)	>1 (no inhibition)	>1 (no inhibition)
CLA + CIP	0.5 (69%)	0.5 (91%)	0.5 (76%)	0.5 (33%)	0.5 (33%)	0.25 (10%)
AMI + CEF	0.25 (73%)	0.25 (91%)	0.5 (92%)	0.125 (12%)	>1	1 (14%)
AMI + CIP	0.5 (72)	0.25 (73%)	1 (88%)	0.5 (30%)	1 (24%)	1 (7%)
CEF + CIP	0.5 (74%)	0.25 (90%)	1 (91%)	1 (20%)	1 (12%)	1 (12%)
CLA + AMI + CEF	0.5 (75%)	0.5 (91%)	1 (90%)	>1 (no inhibition)	>1 (no inhibition)	>1 (no inhibition)
CLA + AMI + CIP	0.5 (74%)	0.5 (92%)	0.5 (66%)	1 (23%)	0.015 (61%)	0.25 (17%)
CLA + CEF + CIP	0.25 (68%)	0.5 (68%)	0.5 (72%)	>1 (no inhibition)	1 (13%)	>1 (no inhibition)
AMI + CEF + CIP	0.25 (73%)	0.25 (91%)	1 (92%)	0.5 (21%)	0.03 (57%)	1 (7%)
4 DRUGS	0.25 (72%)	1 (89%)	1 (90%)	1 (18%)	0.03 (58%)	>1 (no inhibition)

Abbreviations: AMI, amikacin; CEF, cefepime; CIP, ciprofloxacin; CLA, clarithromycin; CSC, cigarette smoke condensate; DS, drug-sensitive; FICI, fractional inhibitory concentration index; MDR, multidrug-resistant; WT, wild-type. Yellow shading, >50% reduction in biofilm density.

## Data Availability

All the datasets generated for this study are included in the article and Appendix A.

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
