# Peer review of "Effects of Anti-Pseudomonal Agents, Individually and in Combination, With or Without Clarithromycin, on Growth and Biofilm Formation by Antibiotic-Susceptible and -Resistant Strains of Pseudomonas aeruginosa, and the Impact of Exposure to Cigarette Smoke Condensate"

_antibiotics, 2025, doi:10.3390/antibiotics14030325_

Round 1
Reviewer 1 Report
Comments and Suggestions for Authors
This study investigated the effects of anti-pseudomonal agents, individually and in combination, with or without clarithromycin, on the growth and biofilm formation of antibiotic-susceptible and resistant strains of Pseudomonas aeruginosa (Psa). In addition, this study investigated the impact of exposure to cigarette smoke condensate (CSC) on these effects which highlights the potential benefits of combining anti-pseudomonal agents with clarithromycin for treating Psa infections, especially in smokers or those exposed to cigarette smoke. This study is designed and organized in a scientific way. The results are soundness and the discussion is helpful for the future research. However, there are some parts could be improved.
- You used Amikacin, cefepime, ciprofloxacin and clarithromycin as the anti-pseudomonal agents, could you explained the reason you chose them? Or is that necessary to test more anti-pseudomonal agents such as Aztreonam, Colistin, Fosfomycin and Rifampin?
- You used three strains of Psa, how about the inclusivity of them? Are they able to cover most of the clinical isolates?
- You mentioned "three separate sets of experiments were conducted in triplicate" in 4.10, could you provide detailed information on the reproducibility of the results across different experimental batches.
- In the reference session, you used varying formats. Some references include full journal names, while others use abbreviations. And some have the doi, but some other don't. Please keep the format consistent.
Reviewer 2 Report
Comments and Suggestions for Authors
The manuscript by Cholo et al. investigated the synergistic effects of amikacin, cefepime, and ciprofloxacin, with and without clarithromycin, against Pseudomonas aeruginosa (Psa) in both planktonic and biofilm states. The study also examined how cigarette smoke condensate (CSC) influences bacterial growth, biofilm formation, and antibiotic interactions.
Overall, this paper demonstrated that the combination of amikacin, cefepime, and ciprofloxacin, particularly with clarithromycin, exhibits synergistic anti-Pseudomonas aeruginosa activity, while cigarette smoke condensate attenuates these effects. However, the authors do not provide sufficient details in the background regarding the significance of their findings. Taking these points and the comments below into consideration, we recommend the manuscript for publication in your journal following major revisions:
- The figure captions should clearly explain what each image (a, b) or table (a, b) represents. For example, a more informative caption would be: "Figure 2. The growth rates of bacterial cultures (a) and biofilm formation (b) of Pseudomonas aeruginosa (Psa) strains compared with the wild-type (WT) at each time point."
- The authors should provide a more detailed discussion of how cigarette smoke condensate (CSC) influences Pseudomonas aeruginosa growth and biofilm formation.
- It would be beneficial for the authors to explain in the introduction why they selected clarithromycin, amikacin, cefepime, and ciprofloxacin for this investigation. Additionally, further justification for the significance of using a macrolide like clarithromycin in combination with anti-pseudomonal agents would strengthen the study's rationale.
- A discussion on the potential clinical implications of this study would be valuable. The authors should elaborate on how their findings could inform treatment strategies for P. aeruginosa infections in smokers.
